# Switch to second-line versus continued first-line antiretroviral therapy for patients with low-level HIV-1 viremia: An open-label randomized controlled trial in Lesotho

Alain Amstutz[1,2,3], Bienvenu Lengo Nsakala[4], Fiona Vanobberghen[1,2], Josephine Muhairwe[4], Tracy Renée Glass[1,2], Tilo Namane[5], Tlali Mpholo[6], Manuel Battegay[2,3], Thomas Klimkait[7], Niklaus Daniel Labhardt[1,2,3]*

1 Swiss Tropical and Public Health Institute, Basel, Switzerland, 2 University of Basel, Basel, Switzerland, 3 Department of Infectious Diseases and Hospital Epidemiology, University Hospital Basel, Basel, Switzerland, 4 SolidarMed, Partnerships for Health, Maseru, Lesotho, 5 Motebang Government Hospital, Leribe, Lesotho, 6 Senkatana HIV Clinic, Maseru, Lesotho, 7 Molecular Virology, Department of Biomedicine, University of Basel, Basel, Switzerland

* n.labhardt@swisstph.ch

**Data Availability Statement:** The key pseudo-anonymised individual participant data, including

## Abstract

### Background

Current World Health Organization (WHO) antiretroviral therapy (ART) guidelines define virologic failure as two consecutive viral load (VL) measurements ≥1,000 copies/mL, triggering empiric switch to next-line ART. This trial assessed if patients with sustained low-level HIV-1 viremia on first-line ART benefit from a switch to second-line treatment.

### Methods and findings

This multicenter, parallel-group, open-label, superiority, randomized controlled trial enrolled patients on first-line ART containing non-nucleoside reverse transcriptase inhibitors (NNRTI) with two consecutive VLs ≥100 copies/mL, with the second VL between 100–999 copies/mL, from eight clinics in Lesotho. Consenting participants were randomly assigned (1:1), stratified by facility, demographic group, and baseline VL, to either switch to second-line ART (switch group) or continued first-line ART (control group; WHO guidelines). The primary endpoint was viral suppression (<50 copies/mL) at 36 weeks. Analyses were by intention to treat, using logistic regression models, adjusted for demographic group and baseline VL. Between August 1, 2017, and August 7, 2019, 137 individuals were screened, of whom 80 were eligible and randomly assigned to switch ($n$ = 40) or control group ($n$ = 40). The majority of participants were female (54 [68%]) with a median age of 42 y (interquartile range [IQR] 35–51), taking tenofovir disoproxil fumarate/lamivudine/efavirenz (49 [61%]) and on ART for a median of 5.9 y (IQR 3.3–8.6). At 36 weeks, 22/40 (55%) participants in the switch versus 10/40 (25%) in the control group achieved viral suppression (adjusted difference 29%, 95% CI 8%–50%, $p$ = 0.009). The switch group had significantly higher probability of viral suppression across different VL thresholds (<20, <100, <200, <400, and <600

primary and secondary endpoints, along with a data dictionary, are available at the time of publication through the data repository Zenodo (DOI 10.5281/zenodo.3953494; https://zenodo.org/record/3953494#.XxbfbkJS-bg). The full data-set can be accessed upon request to the Clinical Statistics and Data Management Group of the Department of Medicine at the Swiss Tropical and Public Health Institute (clin.stats.dm@swisstph.ch) and after signing a data confidentiality agreement.

**Funding:** The study was supported by two grants from the Swiss National Science Foundation (grants IZ07Z0_160876/1 and PCEFP3_181355), obtained by NDL. AA receives his salary through a grant from the MD-PhD programme of the Swiss National Science Foundation (grant 323530_177576). http://www.snf.ch/en/Pages/default.aspx. The funder had no role in study design, data collection and analysis, decision to publish, or preparation of the manuscript.

**Competing interests:** The authors have declared that no competing interests exist.

**Abbreviations:** 3TC, lamivudine; ABC, abacavir; AE, adverse event; ALT, alanine aminotransferase; ART, antiretroviral therapy; AST, aspartate aminotransferase; AZT, zidovudine; BMI, body mass index; BP, blood pressure; CI, confidence interval; CRF, case reporting form; CTCAE, Common Terminology Criteria for Adverse Events; d4T, stavudine; DBS, dried blood spot; eGFR, estimated glomerular filtration rate; EFV, efavirenz; HDL, high-density lipoprotein; IQR, interquartile range; LDL, low-density lipoprotein; LLV, low-level viremia; LTFU, lost to follow-up; NRTI, nucleoside reverse transcriptase inhibitor; NNRTI, non-nucleoside reverse transcriptase inhibitor; NVP, nevirapine; PI, protease inhibitor; SE, standard error; SESOTHO, Switch Either near Suppression Or THOusand; TB, tuberculosis; TC, total cholesterol; TDF, tenofovir disoproxil fumarate; VL, viral load; WHO, World Health Organization.

copies/mL) but not for <1,000 copies/mL. Thirty-four (85%) participants in switch group and 21 (53%) in control group experienced at least one adverse event (AE) ($p = 0.002$). No hospitalization or death or other serious adverse events were observed. Study limitations include a follow-up period too short to observe differences in clinical outcomes, missing values in CD4 cell counts due to national stockout of reagents during the study, and limited generalizability of findings to other than NNRTI-based first-line ART regimens.

## Conclusions

In this study, switching to second-line ART among patients with sustained low-level HIV-1 viremia resulted in a higher proportion of participants with viral suppression. These results endorse lowering the threshold for virologic failure in future WHO guidelines.

## Trial registration

The trial is registered at ClinicalTrials.gov, NCT03088241.

## Author summary

### Why was this study done?

- WHO guidelines define virologic treatment failure as twice a HIV viral load (VL) measurement $\geq$1,000 copies of HIV RNA per milliliter despite good adherence. In contrast, guidelines used in middle- and high-income countries set the threshold for virologic failure and thus need for switch of antiretroviral therapy (ART) at a lower level of viremia.

- Observational studies indicate that persons with sustained unsuppressed viremia below 1,000 copies/mL have worse outcomes. As a result, there is a debate as to whether the WHO should lower the threshold.

- To date, there is no published randomized trial assessing whether a regimen switch at a lower level of viremia is beneficial.

### What did the researchers do and find?

- Our open-label, multicenter, randomized controlled trial conducted in Lesotho (southern Africa) randomized patients taking non-nucleoside reverse transcriptase inhibitor (NNRTI)-based first-line ART with sustained unsuppressed VL 100–999 copies/mL to continued first-line (control group, WHO guidelines) or switch of regimen (switch group).

- Switch to second-line ART resulted in a significantly higher proportion of participants achieving viral suppression below 50 copies/mL at 24 and 36 weeks of follow-up.

- We found no difference in viral rebound above 1,000 copies/mL nor in clinical outcomes between the groups.

- The visual abstract summarizes the trial and its primary endpoint (S1 Fig).

**What do these findings mean?**

- Persons taking first-line NNRTI-based ART with unsuppressed VL between 100 and 999 copies/mL benefit from a regimen switch in terms of viral suppression.

- The findings endorse lowering the WHO VL threshold for treatment failure.

## Introduction

For persons living with HIV, the aim of antiretroviral therapy (ART) is to block viral replication and thus achieve suppression of viremia below the detection limit—usually 20 or 50 copies/mL [1]. A sustained HIV viral load (VL) below the detection limit significantly reduces the risk of AIDS-defining events, severe non-AIDS-defining events and death, and prevents onward transmission [2,3]. Thus, HIV guidelines globally recommend VL measurement as the preferred ART monitoring strategy. However, the threshold for defining virologic failure is subject to debate. Treatment guidelines of resource-rich settings define virologic failure as two consecutive VL ≥200 copies/mL, triggering a switch of ART regimen [1,4,5]. In contrast, World Health Organization (WHO) guidelines that inform most national guidelines in resource-limited settings (in particular, Africa, where approximately 69% of people [26 million] with HIV infection live [6]) set the threshold for virologic failure and thus the need for ART switch at 1,000 copies/mL [7].

Yet several large cohort studies from resource-rich and resource-limited settings observed a higher risk of developing major drug resistance mutations [8,9], virologic failure [10–15], and AIDS-defining events and mortality [16,17] among patients with sustained detectable VL but below 1,000 copies/mL as compared to patients with undetectable VL. Recently, the threshold of 1,000 copies/mL has again been challenged by a large cohort study from South Africa, where patients with unsuppressed VL but below 1,000 copies/mL, usually referred to as low-level viremia (LLV), had up to a five times increased risk of subsequent treatment failure compared to those with undetectable VL [18]. This finding prompted editorialists to urge the WHO to revise the current threshold [19,20]. Until now, to our knowledge, there has been no randomized controlled trial that assessed treatment switch for patients with LLV. As a result, HIV programs following the WHO guidelines usually consider patients with VL <1,000 copies/mL as "virologically suppressed."

In this randomized controlled trial in Lesotho (southern Africa), we assessed if switching participants with VL 100–999 copies/mL to second-line ART would increase the probability of virologic suppression compared to remaining on first-line ART as per WHO guidelines.

## Methods

### Study design and participants

The SESOTHO (Switch Either near Suppression Or THOusand) trial was a parallel-group, open-label, superiority, randomized controlled trial conducted at eight health facilities in four districts of Lesotho (Butha-Buthe, Leribe, Maseru, and Mokhotlong). Recruitment lasted from August 1, 2017, until August 7, 2019, when the target sample size was achieved. A detailed study protocol has been published previously [21]. Eligible participants were individuals living with HIV, taking non-nucleoside reverse transcriptase inhibitor (NNRTI)-based first-line

ART (standard first-line regimen in most resource-limited countries at that time) for at least six mo, presenting two consecutive VLs $\geq$100 copies/mL with the second VL between 100 and 999 copies/mL, and providing written informed consent to participate. Exclusion criteria were poor adherence (self-reported at least 1 dose missed in the last 4 weeks) and active clinical WHO stage 3 or 4 condition at enrollment. The protocol originally planned for recruitment in one district only, but due to slow recruitment, centers in other districts were included, and the protocol amended accordingly. This trial has been approved by the National Health Research and Ethics Committee of the Ministry of Health of Lesotho (ID48-2017, 05.07.2017) and the Ethics committee in Switzerland (Ethikkomission Nordwest- und Zentralschweiz; EKNZ BASEC UBE 2017–00201, 10.04.2017). Prior to randomization, written informed consent was obtained by a study nurse or study physician. Illiterate participants provided a thumbprint, and a witness cosigned the form. For children and young adults aged <18 y, written consent was obtained from a literate caregiver. The informed consent form was written in Sesotho, and the participants received a copy of it.

## Randomization and masking

Eligible and consenting patients were randomized in a 1:1 allocation to switch or control groups using sealed, opaque, and sequentially numbered envelopes. Randomization was stratified by health facility (hospitals; health centers), demographic group (adults; children <16 y; pregnant women) and baseline VL (100–599; 600–999 copies/mL), with randomly varying block sizes of four and six. The randomization list was generated by the trial statistician and the envelopes were prepared by persons independent from the trial. There were two violations of the randomization sequence. First, an investigator once erroneously picked the last instead of the first envelope of the respective stratum at a site. Second, one envelope was prepared by the independent person with the wrong allocation form. Both errors were detected after the participant had completed their baseline visit, and both participants remained in the trial as randomized. Importantly, concealment of allocation was maintained at all times. This was a pragmatic open-label trial, i.e., participants as well as health care providers knew which ART was being administered.

## Procedures and interventions

Participants randomized to the control group remained on first-line ART (standard of care), whereas participants in the switch group were switched to a second-line ART regimen according to the national ART guidelines of Lesotho that follow the WHO consolidated guidelines [7,22]. Regimen switch entailed changing the NNRTI to a protease inhibitor (PI), keeping lamivudine, and changing the other nucleoside reverse transcriptase inhibitor (NRTI) to one that the participant had not previously taken. Drug doses followed the national ART guidelines [22].

At enrollment, the study nurse or physician documented clinical and sociodemographic characteristics on a paper-based case reporting form (CRF). Several preplanned blood tests were conducted: CD4 count using FACSCount or PIMA Alere, alanine aminotransferase (ALT), aspartate aminotransferase (AST), and serum creatinine using Pentra C400, hemoglobin using Yumizen H550, and total cholesterol (TC), high-density lipoprotein (HDL) cholesterol, low-density lipoprotein (LDL) cholesterol, and triglycerides using the Afinion Lipid Panel by Abbott. Additional venous blood was collected, centrifuged and plasma frozen at –80˚ C at Butha-Buthe Government Hospital. These plasma aliquots were shipped to a reference laboratory in Switzerland for amplification and subsequently to Germany for next-generation sequencing (using Illumina MiSeq). Baseline VL was defined as that closest to randomization,

up to 14 weeks before. This window allowed sufficient time for the standard turn-around time of VL results in this setting and to request eligible patients to return for screening. Each participant was followed up for 36 weeks with a blood draw and clinical assessment (including adverse events and self-reported adherence) at 12, 24, and 36 weeks. Laboratory measurements of ALT, AST, hemoglobin, and serum creatinine were performed at all three follow-up study visits, VL at 24 and 36 weeks, and CD4 count and blood lipid profile at 36 weeks. HIV RNA were quantified on a Roche AmpliPrep-Taqman system (AmpliPrep/COBAS TaqMan).

## Outcomes

The primary endpoint was viral suppression defined as VL <50 copies/mL at 36 weeks after randomization. The rationale for this time-point was that, according to WHO and Lesotho national guidelines, VL is measured six mo after switch to second-line ART. If this VL is above 1,000 copies/mL, enhanced adherence counselling is provided and a follow-up VL performed three mo later. Thus, 36 weeks was the earliest possible time point to meet the criteria of virologic failure (versus viral resuppression) after switch to second-line ART.

The secondary endpoints were viral suppression defined by different thresholds (VL <100, <200, <400, <1,000 copies/mL) at 36 weeks, viral suppression (<50 copies/mL) at 24 weeks, sustained virologic failure (detectable VL ≥50 copies/mL at 24 and 36 weeks), self-reported adherence at 12, 24, and 36 weeks, changes in body weight, hemoglobin, CD4 count, and blood lipids from baseline to 36 weeks, new clinical WHO 3 or 4 events or death, and adverse and serious adverse events (graded according to the Common Terminology Criteria for Adverse Events [CTCAE] [23]).

## Data management

All data were double entered into a password-protected database using EpiData (v4.6.0.2). Data integrity checks written into the database limited missing fields and incorrect data entry. Data queries were sent to the local study team for follow-up and correction, as needed. Database closure was on May 23, 2020.

## Statistical methods

Details of our sample size calculation have been published previously [21]. Assuming a two-sided type-1 error of 5% and a power of 90%, 80 individuals (40 per group) were needed to detect a 35% difference in viral suppression. We analyzed outcomes by intention-to-treat. Participants who were switched to next-line ART due to clinical failure prior to 36 weeks or did not have VL measured at 36 weeks were considered to have unsuppressed VL. For categorical outcomes, we used logistic regression models, reporting odds ratios and risk differences with standard errors (SEs) estimated using the delta method [24]. All estimates are reported with 95% confidence intervals (CIs). For continuous outcomes, we used linear regression models. All models were adjusted for demographic group and baseline VL but not health facility (also a stratification factor) in accordance with our statistical analysis plan (S1 Statistical Analysis Plan) due to few participants at some facilities (S1 Table). We performed a number of prespecified sensitivity analyses for the primary outcome: adjustment for baseline covariates with imbalance between groups (determined by the clinical team by visual inspection before seeing outcome data), use of narrower predefined primary endpoint visit windows (S2 Table), and restricting to participants who had VL available and had not changed ART regimen line due to clinical failure (per protocol set). We planned to assess effect modification by demographic group and baseline VL and to perform a subgroup analysis among participants with >0.5 log-drop between screening VLs. However, only four children, no pregnant women, and 13

participants with >0.5 log-drop were enrolled; therefore, it was not possible to perform analyses in these groups. The VL suppression threshold of <20 copies/mL was added as a post hoc analysis since this is the lower level of detection of the VL system in Lesotho (AmpliPrep/COBAS TaqMan), and the threshold of <600 copies/mL was added further to a reviewer request to correspond to the VL stratification used at enrollment. As a post hoc analysis, we analyzed the primary endpoint in the subgroup of those participants with a baseline VL 200 to 999 copies/mL because most guidelines from resource-rich settings use a VL threshold of 200 copies/mL to define virologic failure. Good adherence was defined as self-report of no missed doses in the previous four weeks [25,26]. Laboratory safety parameters were assessed according to CTCAE [23]. Visit windows of 8–<20, 20–<32 and 32–<52 weeks were permitted for the 12-, 24- and 36-week visits, respectively. Analyses were performed using Stata (StataCorp. 2017. *Version 15*). No adjustments were made for multiple testing. Clinical, adherence and laboratory outcomes were assessed among those with non-missing data. The trial is registered with ClinicalTrials.gov, number NCT03088241, and the CONSORT checklist is provided (S1 CONSORT checklist).

## Results

Between August 1, 2017, and August 7, 2019, 137 patients were screened, of whom 80 (58%) were eligible and randomized to the switch (*n* = 40 [50%]) or control group (*n* = 40 [50%]) (Fig 1). Clinical and biomedical baseline characteristics are presented in Table 1 and sociodemographic characteristics in S3 Table. The median age was 42 y (interquartile range [IQR] 35–51) and 54 (68%) were female. One participant in the switch and three participants in the control group were children. Seventy-nine participants (99%) were clinical WHO stage 1; one participant was WHO stage 2. Participants had been on ART for a median of 5.9 y (IQR 3.3–8.6), the most common ART regimen was tenofovir disoproxil fumarate/lamivudine/efavirenz (*n* = 49, 61%), and none had been exposed to antiretroviral classes other than nucleoside reverse transcriptase inhibitors (NRTIs) and NNRTIs. Median baseline CD4 count was 436 cells/uL (IQR 293–702). Median baseline VL was 347 copies/mL (IQR 170–648), with 56 (70%) participants between 100 and 599 copies/mL and 24 (30%) between 600 and 999 copies/mL. Baseline characteristics were balanced between groups, except for greater use of cotrimoxazole prophylaxis and other comedication, lower CD4 count, and higher alcohol use in the switch compared to control group, as well as some differences in access to their health facility (Table 1, S3 Table). Due to the low level of viremia, genotypic resistance testing was successful in only 29 (36%) participants (15 in switch and 14 in control group; S4 Table). Among these, 25 (86%) harbored HIV with Stanford level 5 resistance against at least two drugs in their current ART regimen.

Following randomization, all participants in the control group remained on the same first-line regimen as before enrollment, and all participants in the switch group switched to second-line ART (ART regimens detailed in S5 Table). By 36 weeks, one participant in the switch group had requested to change back to their first-line regimen (two days after enrollment), and three participants in the control group had switched to second-line ART (one due to treating physician's decision upon immunological failure and >10% weight loss, and two due to stock out of their first-line regimen and national transition to dolutegravir-based first-line ART).

For the primary outcome at 36 weeks, VL results were available for 39 (98%) and 37 (93%) participants in the switch and control groups, respectively (Fig 1). Twenty-two out of 40 (55%) and 10 out of 40 (25%) participants in the switch and control groups, respectively, achieved viral suppression (Table 2), yielding an adjusted absolute difference of 29% (95% CI 8 to 50;

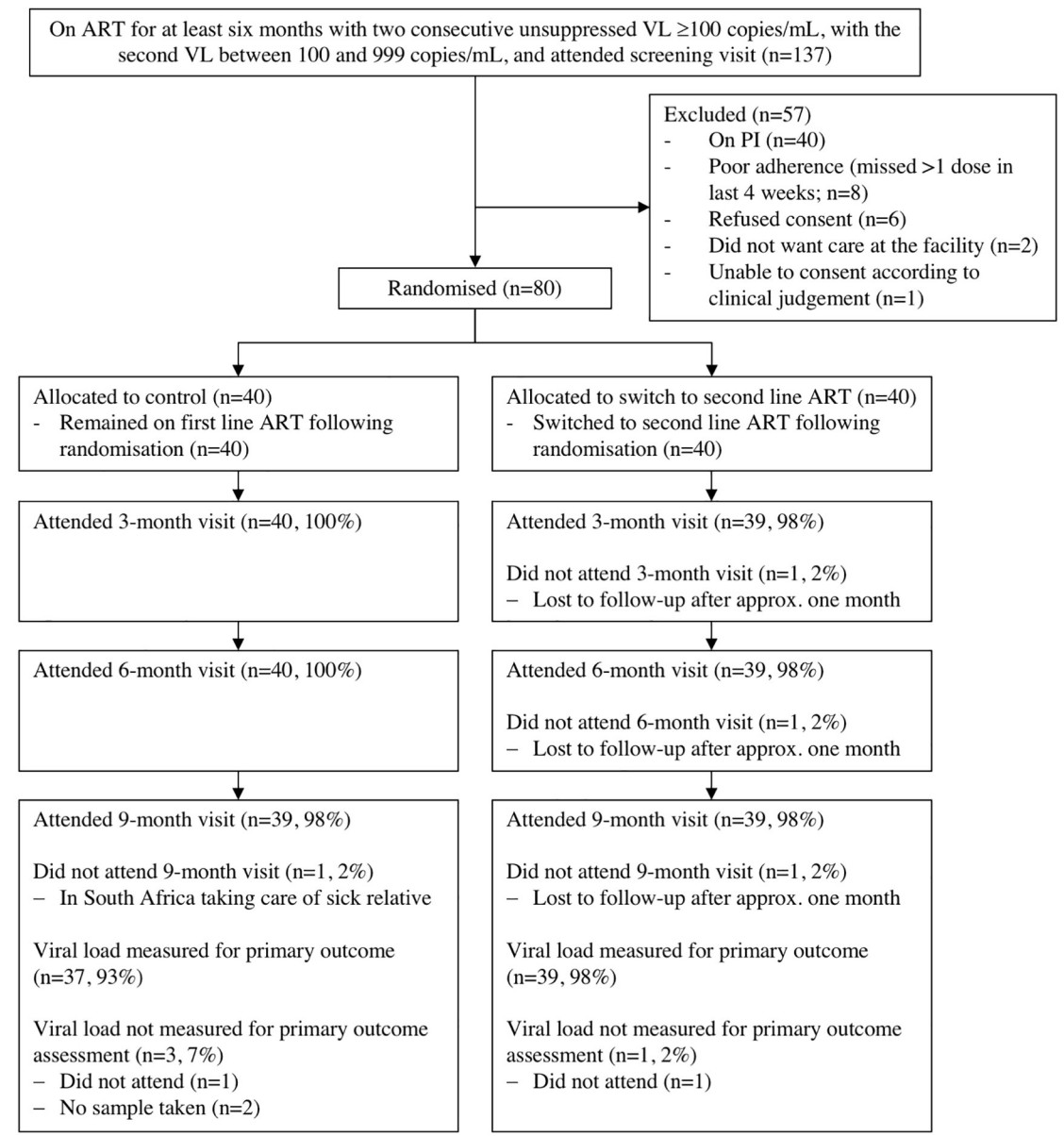

**Fig 1. Participant flow diagram of SESOTHO trial.** ART, antiretroviral therapy; LTFU, lost to follow-up; PI, protease inhibitor; SESOTHO, Switch Either near Suppression Or THOusand; VL, viral load

$p$ = 0.009). Results were robust to sensitivity analyses (S2 Table). The intervention effect among those with baseline VL 600–999 copies/mL (9/13 [69%] in the switch and 2/11 [18%] in the control group; adjusted risk difference 48% [95% CI 21 to 76]) was nonsignificantly greater than the effect among those with baseline VL 100–599 copies/mL (12/27 [48%] in the switch and 8/29 [28%] in the control group; adjusted risk difference 18% (95% CI −5 to 42); interaction $p$ = 0.20). In a post hoc analysis, among the 55 participants with baseline VL 200–999 copies/mL, 14/26 (54%) in the switch versus 5/29 (17%) in the control group achieved viral suppression (adjusted risk difference: 34%, 95% CI 10 to 58; $p$ = 0.009).

At 36 weeks, larger proportions of participants in the switch versus control groups achieved VL <20, <100, <200, <400, and <600 copies/mL but not VL <1,000 copies/mL (Table 2).

**Table 1. Clinical and biomedical baseline characteristics of trial participants.**

| | Control group (n = 40) | Switch group (n = 40) | Total (n = 80) |
|---|---|---|---|
| Child[a] | 3 (8%) | 1 (3%) | 4 (5%) |
| Age, y | 42 (36–52) [5–75] | 42 (34–50) [13–73] | 42 (35–51) [5–75] |
| Sex, female | 27 (68%) | 27 (68%) | 54 (68%) |
| Weight adults, kg | 64 (58–84) [44–127] | 67 (50–80) [39–109] | 67 (57–81) [39–127] |
| BMI adults, kg/m$^2$ [b] | 26 (22–31) [18–51] | 24 (21–30) [17–46] | 25 (21–30) [17–51] |
| Underweight (<18.5) | 1 (3%) | 3 (8%) | 4 (6%) |
| Normal (18.5–<25) | 14 (40%) | 18 (49%) | 32 (44%) |
| Overweight/obese (≥25) | 20 (57%) | 16 (43%) | 36 (50%) |
| Blood Pressure, systolic, mmHg[c] | 120 (110–130) [94–172] | 110 (110–124) [90–190] | 115 (110–130) [90–190] |
| Blood Pressure, diastolic, mmHg[c] | 79 (68–90) [59–99] | 74 (70–86) [60–132] | 76 (70–87) [59–132] |
| Weight children, kg | 16, 20, 25 | 33 | 16, 20, 25, 33 |
| Current clinical WHO stage 1 | 39 (98%) | 40 (100%) | 79 (99%) |
| Time since ART initiation, y | 5.3 (3.0–8.6) [0.9–12.1] | 6.3 (3.6–8.3) [1.0–13.2] | 5.9 (3.3–8.6) [0.9–13.2] |
| Current ART regimen at screening | | | |
| TDF/3TC/EFV | 22 (58%) | 26 (65%) | 49 (61%) |
| ABC/3TC/EFV | 3 (8%) | 4 (10%) | 7 (9%) |
| AZT/3TC/EFV | 6 (15%) | 3 (8%) | 9 (11%) |
| TDF/3TC/NVP | 4 (10%) | 3 (8%) | 7 (9%) |
| AZT/3TC/NVP | 4 (10%) | 4 (10%) | 8 (10%) |
| On tuberculosis preventive therapy (Isoniazid) | 1 (3%) | 1 (3%) | 2 (3%) |
| On prophylactic co-trimoxazole treatment | 7 (18%) | 13 (33%) | 20 (25%) |
| Other concomitant treatment | 7 (18%) | 14 (35%) | 21 (26%) |
| Alcohol use[d] | 4 (11%) | 8 (21%) | 12 (16%) |
| Illicit use of local cannabis[e] | 2 (5%) | 4 (11%) | 6 (8%) |
| History of TB[f] | 13 (34%) | 16 (40%) | 29 (37%) |
| ART history; ever exposed to: | | | |
| TDF | 27 (68%) | 30 (75%) | 57 (71%) |
| ABC | 3 (8%) | 4 (10%) | 7 (9%) |
| AZT | 10 (25%) | 9 (23%) | 19 (24%) |
| 3TC | 40 (100%) | 40 (100%) | 80 (100%) |
| d4T | 4 (10%) | 7 (18%) | 11 (14%) |
| EFV | 32 (80%) | 33 (83%) | 65 (81%) |
| NVP | 9 (23%) | 10 (25%) | 19 (24%) |
| HIV VL, copies/mL | 391 (190–638) [113–990] | 300 (125–679) [102–950] | 347 (170–648) [102–990] |
| HIV VL 100–599 copies/mL | 29 (73%) | 27 (68%) | 56 (70%) |
| HIV VL 600–999 copies/mL | 11 (28%) | 13 (33%) | 24 (30%) |
| CD4 count, cells/mm$^3$ [g] | 478 (369–699) [139–1,237] | 328 (236–704) [129–1,192] | 436 (293–702)] 129–1,237 |
| Hemoglobin, g/dL[h] | 14 (13–15) [10–17] | 13 (13–14) [7–18] | 14 (13–15) [7–18] |
| Creatinine, micromol/L[i] | 67 (59–84) [40–125] | 68 (57–81) [27–191] | 68 (58–81) [27–191] |
| eGFR, mL/min[i,j] | 104 (81–123) [46–194] | 99 (81–127) [41–185] | 101 (81–127) [41–194] |
| ALT, U/L[k] | 23 (18–33) [6–63] | 27 (19–33) [10–117] | 25 (19–33) [6–117] |
| AST, U/L[l] | 27 (23–33) [8–58] | 29 (26–37) [15–124] | 28 (24–35) [8–124] |
| Total cholesterol, mmol/L[m] | 4.1 (3.5–4.7) [2.7–7.3] | 4.3 (3.7–5.1) [2.6–6.6] | 4.2 (3.5–4.9) [2.6–7.3] |
| HDL, mmol/L[n] | 1.3 (1.2–1.4) [0.9–1.9] | 1.4 (1.1–1.6) [0.9–2.1] | 1.4 (1.2–1.6) [0.9–2.1] |
| LDL, mmol/L[o] | 2.3 (1.6–2.8) [0.8–4.6] | 2.3 (1.7–3.1) [1.3–4.1] | 2.3 (1.7–3.0) [0.8–4.6] |
| Triglycerides, mmol/L[p] | 1.1 (0.8–1.6) [0.6–7.0] | 1.0 (0.7–1.7) [0.5–2.2] | 1.0 (0.8–1.7) [0.5–7.0] |

(*Continued*)

**Table 1.** (Continued)

| | Control group (n = 40) | Switch group (n = 40) | Total (n = 80) |
|---|---|---|---|
| Total/HDL cholesterol ratio[n] | 3.3 (2.6–3.8) [1.9–4.7] | 2.9 (2.6–3.9) [2.0–5.9] | 3.0 (2.6–3.9) [1.9–5.9] |

Results are number (% of those with nonmissing data) for categorical variables and median (IQR) [range] for continuous variables.

a Defined as <16 y old

b Missing for 2 participants in control group and 2 participants in switch group, due to height never having been measured

c Missing for 2 participants in control group and 2 participants in switch group

d Female: at least 2 drinks every day or regularly 4 or more drinks on one occasion; Male: at least 3 drinks every day or regularly 5 or more drinks on one occasion

e Missing for one participant in switch group

f Missing for 2 participants in control group

g Not measured for 40 participants (21 in control group and 19 in switch group)

h Not measured for 9 participants (6 in control group and 3 in switch group)

i Not measured for 9 participants (4 in control group and 5 in switch group)

j Estimated using Cockcroft-Gault equation

k Not measured for 16 participants (8 in control group and 8 in switch group)

l Not measured for 17 participants (9 in control group and 8 in switch group)

m Not measured for 24 participants (12 in control group and 12 in switch group)

n Not measured for 34 participants (18 in control group and 16 in switch group)

o Not measured for 37 participants (20 in control group and 17 in switch group)

p Not measured for 29 participants (15 in control group and 14 in switch group)

Abbreviations: 3TC, lamivudine; ABC, abacavir; ALT, alanine aminotransferase; ART, antiretroviral therapy; AST, aspartate aminotransferase; AZT, zidovudine; BMI, body mass index; d4T, stavudine; eGFR, estimated glomerular filtration rate; EFV, efavirenz; HDL, high-density lipoprotein; IQR, interquartile range; LDL, low-density lipoprotein; NVP, nevirapine; TB, tuberculosis; TDF, tenofovir; VL, viral load; WHO, World Health Organization.

**Table 2. Primary and secondary virologic outcomes.**

| | Overall (n = 80) | Control group (n = 40) | Switch group (n = 40) | Odds ratio (95% CI)[a] | Risk difference[a,e] | P-value[a] |
|---|---|---|---|---|---|---|
| Primary endpoint: VL <50 copies/mL at 36 weeks | 32 (40%) | 10 (25%) | 22 (55%) | 3.55 (1.37,9.24) | 29 (8,50) | 0.009 |
| VL <50 copies/mL at 24 weeks[b] | 26 (33%) | 6 (15%) | 20 (50%) | 5.70 (1.91,17.0) | 34 (15,53) | 0.002 |
| VL at 36 weeks[c] | | | | | | |
| VL <20 copies/mL | 28 (35%) | 7 (18%) | 21 (53%) | 5.14 (1.82,14.6) | 34 (15,54) | 0.002 |
| VL <100 copies/mL | 43 (54%) | 17 (43%) | 26 (65%) | 2.50 (1.00,6.21) | 22 (1,44) | 0.05 |
| VL <200 copies/mL | 51 (64%) | 19 (48%) | 32 (80%) | 4.73 (1.73,13.0) | 34 (14,53) | 0.003 |
| VL <400 copies/mL | 58 (73%) | 24 (60%) | 34 (85%) | 3.85 (1.30,11.4) | 25 (6,44) | 0.02 |
| VL <600 copies/mL | 60 (75%) | 26 (65%) | 34 (85%) | 3.11 (1.04,9.27) | 20 (2,39) | 0.04 |
| VL <1,000 copies/mL | 66 (83%) | 32 (80%) | 34 (85%) | 1.44 (0.45,4.64) | 5 (-11,22) | 0.54[d] |

a Estimated by logistic regression adjusted for demographic group and baseline VL.

b Of the 54 participants who did not achieve VL<50 copies/mL at 24 weeks, 11 did not have VL measured (4 in control group and 7 in switch group), and 1 in the control group had switched to second line due to clinical failure.

c For each of the thresholds indicated, 4 participants were considered not to meet the threshold because they did not have VL measured (1 in control group and 3 in switch group), and 1 in the control group had switched to second line due to clinical failure.

d Adjusted for baseline VL only since all four children achieved VL <1,000 copies/mL at 36 weeks. Instead fitting the model among the 76 adults only yields an OR of 1.55 (0.48,5.01), risk difference of 6 (-11,24), p-value 0.47.

e Confidence intervals estimated using delta method

Abbreviations: CI, confidence interval; VL, viral load.

The number of participants with sustained virologic failure (detectable VL $\geq$50 copies/mL at 24 and 36 weeks) was 12/40 (30%) and 28/40 (70%) in the switch and control groups, respectively (adjusted odds ratio 0.18 [95% CI 0.07 to 0.48]; risk difference −40% [95% CI −60 to −20]; $p$ = 0.001). Overall, 48 (62%) participants had self-reported good adherence at every follow-up visit (20 [51%] in the switch and 28 [72%] in the control group; adjusted odds ratio 0.43 [95% CI 0.17 to 1.10]; risk difference −20% [95% CI −41 to 2]; $p$ = 0.08).

There were 8 (10%) participants with >10% weight loss (4 in each group); no other WHO stage 3 or 4 events were reported. There were no deaths or hospitalizations. Due to repeated national stock-outs in reagents, only 29 (36%) participants had CD4 count at baseline and 36 weeks. For similar reasons, complete blood-lipid profile at baseline and follow-up is available for 59 (74%) participants only. There was no evidence of a difference in changes in mean body weight, hemoglobin, CD4 count, and blood lipid levels between baseline and 36 weeks, apart from a slightly larger increase in triglycerides and total/HDL cholesterol ratio in the switch versus control groups, although these results should be interpreted with caution due to the large amount of missing data (Table 3).

There were 102 AEs reported in 55 (69%) participants, with a greater proportion of participants in the switch group experiencing at least one AE compared to those in the control group (34 [85%] versus 21 [53%], $p$ = 0.002; S6 Table). Overall, 41 (40%) AEs were judged as may be

**Table 3. Change in clinical parameters at 36 weeks versus baseline.**

| | Overall ($n$ = 80) | Control group ($n$ = 40) | Switch group ($n$ = 40) | Mean difference (95% CI)[a] | $P$-value[a] |
|---|---|---|---|---|---|
| Mean change in body weight, kg (SE)[b] | 0.35 (0.50) | 1.07 (0.77) | −0.33 (0.65) | −1.45 (−3.48,0.59) | 0.16 |
| Mean BMI, change, kg/m2 (SE)[c,d] | 0.08 (0.20) | 0.32 (0.32) | −0.14 (0.26) | −0.54 (−1.36,0.28) | 0.19 |
| Mean hemoglobin change, g/dL (SE)[e] | −0.17 (0.18) | −0.03 (0.15) | −0.30 (0.33) | −0.40 (−1.14,0.33) | 0.28 |
| Mean CD4 count, cells/mm³ (SE)[f,g] | 35.14 (47.11) | −5.67 (58.69) | 78.86 (75.13) | 97.91 (−64.95,260.77) | 0.23 |
| Mean change in total cholesterol, mmol/L (SE)[h] | 0.11 (0.14) | 0.08 (0.18) | 0.14 (0.23) | 0.10 (−0.52,0.72) | 0.74 |
| Mean change in HDL cholesterol, mmol/L (SE)[i] | −0.07 (0.04) | −0.02 (0.06) | −0.16 (0.06) | −0.09 (−0.27,0.09) | 0.30 |
| Mean change in LDL cholesterol, mmol/L (SE)[j] | 0.23 (0.11) | 0.15 (0.12) | 0.34 (0.20) | 0.20 (−0.31,0.71) | 0.43 |
| Mean change in triglycerides, mmol/L (SE)[k] | 0.21 (0.10) | 0.13 (0.13) | 0.32 (0.13) | 0.29 (0.01,0.58) | 0.04 |
| Mean change in Total/HDL cholesterol ratio (SE)[i] | 0.43 (0.11) | 0.25 (0.14) | 0.72 (0.12) | 0.43 (0.06,0.80) | 0.02 |
| Mean change in Systolic BP, mmHg (SE)[c,l] | −3.73 (1.99) | −2.20 (2.46) | −5.31 (3.17) | −3.61 (−9.48,2.26) | 0.22 |
| Mean change in Diastolic BP, mmHg (SE)[c,l] | −2.32 (1.67) | 0.23 (2.26) | −4.97 (2.41) | −3.82 (−8.86,1.22) | 0.13 |

a Estimated by linear regression adjusted for demographic group, baseline VL and the baseline value of each respective parameter. No transformations were applied.

b Complete data available for 39/40 (98%) in switch and 37/40 (93%) in control group.

c Among adults only.

d Complete data available for 36/40 (90%) in switch and 33/40 (83%) in control group.

e Complete data available for 34/40 (85%) in switch and 32/40 (80%) in control group.

f Complete data available for 14/40 (35%) in switch and 15/40 (38%) in control group.

g Large standard errors driven by a few participants with large changes in CD4 count between baseline and 36 weeks, which were checked and confirmed correct (7 participants in control group had absolute changes >100 cells/mm³, of which one dropped from 1237 to 609 cells/mm³; and 9 participants in intervention group had absolute changes >100 cells/mm³, of which two increased from around 200–300 cells/mm³ to 800–900 cells/mm³).

h Complete data available for 18/40 (45%) in switch and 24/40 (60%) in control group.

i Complete data available for 13/40 (33%) in switch and 20/40 (50%) in control group.

j Complete data available for 13/40 (33%) in switch and 18/40 (45%) in control group.

k Complete data available for 15/40 (38%) in switch and 22/40 (55%) in control group.

l Complete data available for 29/40 (73%) in switch and 30/40 (75%) in control group.

Abbreviations: BMI, body mass index; BP, blood pressure; CI, confidence interval; HDL, high-density lipoprotein; LDL, low-density lipoprotein; SE, standard error; VL, viral load.

related ($n$ = 32) or related ($n$ = 9) to treatment, with more in the switch versus control group (27/40 [68%] versus 8/40 [20%], $p < 0.001$). This was mainly due to a larger number of gastrointestinal AEs in the switch group (S7 Table). Only two (2%) participants experienced a grade 3 AE (one unspecified clinical deterioration in the control group and one renal failure with serum creatinine of 273 micromol/mL). Both resolved at follow-up. There were no grade 4 nor serious AEs. Laboratory safety monitoring revealed one further grade 3 event due to low hemoglobin.

## Discussion

To our knowledge, the SESOTHO trial is the first randomized controlled trial that addresses ART switch in persons with HIV-1 LLV. We observed that switching from NNRTI-based first-line to PI-based second-line ART among patients with persistent LLV (between 100 and 999 copies/mL) increased viral suppression (<50 copies/mL) at 36 weeks compared to continued first-line ART. Moreover, participants in the switch group were more likely to achieve viral suppression already by 24 weeks and sustain it through 36 weeks.

The current WHO consolidated ART guidelines define virologic failure as two consecutive VL ≥1,000 copies/mL despite good adherence, triggering empiric switch to next-line ART. These guidelines imply that patients with LLV are considered as having a "suppressed" viremia and will not receive more intensive VL monitoring nor switch of treatment regimen. Most national HIV programs in sub-Saharan Africa strictly follow these guidelines. Meanwhile, several large cohort studies from resource-rich settings [8–14,16,17], western and eastern Africa [15], and South Africa [18] conclude high risk of virologic failure, AIDS-defining events, emergence of major drug resistance, and increased mortality among patients with sustained LLV. Moreover, in resource-limited settings with less-frequent virologic monitoring and high rates of loss to follow-up (LTFU), one may argue that the use of the lowest possible sensitive treatment failure threshold is important to be able to react in time, compared to resource-rich settings with frequent and reliable laboratory monitoring.

To our knowledge, no randomized controlled trials—neither in resource-rich nor resource-limited settings—have been conducted to inform treatment guidelines on switching to second-line ART among patients with LLV. A review of the literature revealed three retrospective case-control analyses from observational cohort studies that specifically assessed continued ART versus ART regimen switch or intensification among patients experiencing LLV. In a study from Taiwan, among 165 participants with persistent LLV, 46 were switched to second-line and were more likely to achieve viral suppression <40 copies/mL at 48 weeks compared to remaining on first-line ART (82.6% versus 63.0%) [27]. A study from the United States of America analyzed 149 participants with at least two consecutive VLs between 50 and 1,000 copies/mL, of which 26 eventually switched ART and 123 continued with unchanged ART [28]. At six mo, a lower proportion in the switch than the control group achieved viral suppression <100 copies/mL, but a higher proportion in the control than the switch group experienced viral rebound to >1,000 copies/mL. Neither difference was statistically significant. Among 21 participants with two consecutive VLs of 50–500 copies/mL in an Italian cohort, a higher proportion of the 12 participants who switched regimens achieved VL <50 copies/mL compared to those who did not switch [29].

In our trial, among participants with successful genotypic HIV sequencing, 86% harbored HIV with major drug resistance mutations against at least two drugs at baseline. This is in line with an earlier study from Lesotho where adults and children with sustained detectable VL below 1,000 copies/mL were as likely to harbor resistant HIV strains as were those with VL ≥1,000 copies/mL [30]. In this current study, we found higher proportions of viral suppression

in the switch group when considering virologic outcomes at 36 weeks other than <50 copies/mL, i.e., 20, 100, 200, 400, and 600 copies/mL, but not for the current WHO-recommended threshold of 1,000 copies/mL. Worsening of LLV in terms of a viral rebound above 1,000 copies/mL at 36 weeks was rare (17%) and similar in both groups. Furthermore, we observed few AIDS-defining events in either group. The relatively short follow-up period of 36 weeks may explain why we did not observe more cases of clinical failure and viral rebound above 1,000 copies/mL in the control group.

Self-reported drug-adherence was lower in the switch than control group, with only 51% reporting good adherence throughout follow-up in the switch compared to 72% in the control group. This may have been caused by the higher pill burden of the lopinavir/ritonavir-based second-line regimens that consisted of 5–6 pills per day or by the higher occurrence of AEs. Nevertheless, 55% of participants in the switch group achieved viral suppression below 50 copies/mL and 85% below 400 copies/mL. This is in line with findings from the EARNEST trial, which demonstrated high effectiveness of lopinavir/ritonavir-containing second-line regimens in resource-limited settings, even in case of suboptimal adherence or inactive NRTI backbones [31]. Adherence assessments are difficult to compare across different trials, but the overall self-reported adherence in our trial appears to be lower than in the lopinavir/ritonavir treatment arms in two large second-line trials in Africa [26,32]. One reason may be that LLV was less perceived as an important problem among providers as well as patients, and thus, efforts to improve adherence were less intensive. Apart from a higher number of the well-known gastrointestinal AEs of lopinavir/ritonavir among the participants randomized to switch compared to those in the control group, second-line regimens were tolerated well.

Overall, these findings endorse a lower VL threshold in future WHO guidelines. However, higher rates of viral suppression should be balanced against higher rates of reported AEs, particularly gastrointestinal side effects due to lopinavir/ritonavir and their potential long-term consequences to adherence. Furthermore, the results should not only be assessed from an individual perspective but also from a programmatic perspective. First, many HIV programs in rural Africa rely on dried blood spot (DBS) technology to obtain VL measurements. The lower level of VL detection using DBS is traditionally limited to around 1,000 copies/mL [33], although some recent publications indicate a lower possible limit [34] and a new DBS method with promising results for LLV [35]. Second, a lower VL threshold for treatment failure would inevitably lead to more patients needing follow-up VL testing, adherence counselling, and ART regimen switches. It is difficult to number precisely the amount of additional workload caused by lower VL thresholds. A recent cohort study from South Africa observed that over a period of 10 y among 69,454 patients on first-line ART, 16,013 (23%) had a single instance of LLV (51–999 copies/mL), and among these, 1,605 (10%) showed LLV at two consecutive measurements [18]. These data suggest that particularly following up all first-time LLV would require a substantial amount of resources. This may pose an additional burden to already overstretched clinic staff, laboratories, national HIV programs, and budgets. Already using a threshold of 1,000 copies/mL, many national programs struggle to provide timely adherence support, follow-up VL, and switch to second-line ART regimens [36,37]. Conversely, with the large-scale introduction of dolutegravir-containing first-line ART regimens in sub-Saharan Africa, overall higher suppression rates can be expected, counterbalancing the additional burden imposed if the threshold is lowered. More research about the public health implications and cost-effectiveness of lowering the threshold in resource-limited settings is needed.

This study has several limitations. First, it was an open-label randomized trial, where participants and providers were aware of treatment group allocation, for logistical and pragmatic reasons. Second, we powered the trial to assess viral suppression at 36 weeks as an indicator of treatment success. We observed few clinical events and no differences in most clinical

parameters nor viral rebound above 1,000 copies/mL between the groups. However, these assessments were likely underpowered, and the follow-up period too short. Third, due to a national stockout of CD4 reagents during the study period, data on immunological outcomes are too incomplete to draw any conclusions. Fourth, only patients taking an NNRTI-based first-line regimen were eligible. With dolutegravir-containing first-line ART regimens being introduced globally, treatment failure is expected to be less frequent, and unsuppressed VL may more often be caused by poor ongoing adherence than emerging drug resistance [38]. Finally, it is noteworthy that the trial population had been taking first-line ART for a median duration of 5.9 y. Our findings may thus not be applicable to persons with LLV less than one year after ART initiation.

To our knowledge, this is the first randomized controlled trial to assess the effect of switching to second-line ART among persons living with HIV who have LLV on NNRTI-based first-line ART. In summary, switching patients with persistent LLV to second-line ART results in better virologic outcomes compared to continued first-line ART. Our findings endorse lowering the threshold for virologic treatment failure in future WHO guidelines.

## Supporting information

**S1 CONSORT Checklist.**
(DOC)

**S1 Statistical Analysis Plan.**
(PDF)

**S1 Fig. Visual abstract.**
(JPG)

**S1 Table. Enrollment by site.**
(DOCX)

**S2 Table. Sensitivity analyses for the primary endpoint.**
(DOCX)

**S3 Table. Sociodemographic characteristics of trial participants.**
(DOCX)

**S4 Table. Baseline genotypic resistance testing.**
(DOCX)

**S5 Table. ART regimens over time.** ART, antiretroviral therapy.
(DOCX)

**S6 Table. Adverse events—summary.**
(DOCX)

**S7 Table. Adverse events—list.**
(DOCX)

## Acknowledgments

We would like to recognize the hard work and valuable contributions of the study staff at all study sites, the tireless support of the SolidarMed team in Lesotho, as well as the District Health Management Teams. We gratefully acknowledge the trial participants.

## Author Contributions

**Conceptualization:** Alain Amstutz, Tracy Renée Glass, Manuel Battegay, Niklaus Daniel Labhardt.

**Data curation:** Alain Amstutz, Bienvenu Lengo Nsakala, Fiona Vanobberghen.

**Formal analysis:** Fiona Vanobberghen.

**Funding acquisition:** Niklaus Daniel Labhardt.

**Investigation:** Alain Amstutz, Bienvenu Lengo Nsakala, Josephine Muhairwe, Tilo Namane, Tlali Mpholo, Thomas Klimkait, Niklaus Daniel Labhardt.

**Methodology:** Alain Amstutz, Fiona Vanobberghen, Tracy Renée Glass, Thomas Klimkait, Niklaus Daniel Labhardt.

**Project administration:** Bienvenu Lengo Nsakala, Josephine Muhairwe, Niklaus Daniel Labhardt.

**Supervision:** Niklaus Daniel Labhardt.

**Visualization:** Alain Amstutz, Niklaus Daniel Labhardt.

**Writing – original draft:** Alain Amstutz, Niklaus Daniel Labhardt.

**Writing – review & editing:** Alain Amstutz, Fiona Vanobberghen, Josephine Muhairwe, Tracy Renée Glass, Tilo Namane, Tlali Mpholo, Manuel Battegay, Thomas Klimkait, Niklaus Daniel Labhardt.

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
