## [Editor Report · Decision Letter 0]

17 Jun 2020

Dear Dr Labhardt, 

Thank you for submitting your manuscript entitled "Switch to second-line versus continued first-line antiretroviral therapy for patients with low-level HIV-1 viremia – an open-label randomized controlled trial" for consideration by PLOS Medicine.

Your manuscript has now been evaluated by the PLOS Medicine editorial staff and I am writing to let you know that we would like to send your submission out for external assessment.

Kind regards,

Richard Turner, PhD

Senior editor, PLOS Medicine

rturner@plos.org

---

## [Decision Letter · Decision Letter 1]

3 Jul 2020

Dear Dr. Labhardt,

Thank you very much for submitting your manuscript "Switch to second-line versus continued first-line antiretroviral therapy for patients with low-level HIV-1 viremia – an open-label randomized controlled trial" (PMEDICINE-D-20-02716R1) for consideration at PLOS Medicine. 

Your paper was evaluated by the in-house editors and sent to independent reviewers, including a statistical reviewer. The reviews are appended at the bottom of this email and any accompanying reviewer attachments can be seen via the link below:

[LINK]

In light of these reviews, we will not be able to accept the manuscript for publication in the journal in its current form, but we would like to invite you to submit a revised version that addresses the reviewers' and editors' comments fully. You will appreciate that we cannot make a decision about publication until we have seen the revised manuscript and your response, and we expect to seek re-review by one or more of the reviewers. 

We hope to receive your revised manuscript by Jul 24 2020 11:59PM. Please email us (plosmedicine@plos.org) if you have any questions or concerns.

Please let me know if you have any questions. Otherwise, we look forward to receiving your revised manuscript in due course. 

Sincerely,

Richard Turner, PhD

rturner@plos.org

To your data statement, please add non-author contact details for those wishing to inquire about access to the full dataset.

Please quote the setting in your title, and substitute a colon before the study descriptor. 

We ask you to add a sentence to the abstract to quote the higher proportion of "switch" participants experiencing adverse events (as discussed around line 356). 

Please add a new final sentence to the "methods and findings" subsection of your abstract, beginning "Study limitations include ..." or similar and quoting 2-3 of the study's main limitations. 

Please add "In this study ..." or similar at line 65. 

At line 284 you use the word "trend". Please make it clear whether this refers to a statistical trend (the relevant p value appears to be 0.2), and we suggest adapting the wording to the form "X was non-significantly greater than Y" if appropriate. 

Please revisit the first paragraph of your discussion section, which should summarize the study's findings. Rather than "... we have demonstrated ... increases viral suppression", we ask you to report the study's findings in the paste tense (e.g., "... we observed that ... increased ...").

Where you make a claim of "the first", as at line 364 for example, please add "to our knowledge" or similar. 

Please move reference call-outs before punctuation throughout the paper, e.g. "... onward transmission [2,3].".

Please remove trademarks from the paper. 

Please remove the information on data sharing, funding and conflict of interest from the end of the text. This information will appear in the metadata in the event of publication, via information provided in the submission form. 

Please revisit your reference list to ensure that all citations meet journal format. All italics should be converted into plain text; six author names should be listed rather than 3, followed by "et al." where appropriate; and for reference 3, for example, "Lancet" will suffice for the journal name. 

Please rename figure 1 "Participant flow diagram" or similar. 

We ask that you attach a completed CONSORT checklist for your study, referred to in the methods section ("see S1_CONSORT_Checklist" or similar). In the checklist, please refer to individual items by section (e.g., "Methods") and paragraph numbers rather than by page or line numbers, as the latter generally change in the event of publication. 

Comments from the reviewers:

*** Reviewer #1: 

In this trial, participants with a VL >100 were switched, or not, and 36 weeks later their chances of being <50 was assessed. 

The failure threshold of <1000 was chosen based on transmission risk. The authors need to explain the negative clinical implications of low level viraemia beyond simply having a greater risk of viral failure. People who were switched earlier experienced more toxicity and worse adherence, so the clinical benefits need to be made clear. The fact that low level viraemia is a risk factor for viral failure is logical, just as being aged 49 is a "risk factor" for turning 50. 

Why was 36 weeks chosen for time to suppression?

 Line 101: "Viral suppression should be achieved by latest six months after ART initiation". This depends on viral load at initiation and regimen.

Line 192: What stats programme was used?

Line 215: Why was self-report of no missed doses in the previous four weeks chosen to define "good adherence"

Line 223: It would be helpful if the viral load stratifications in the baseline characteristics and outcomes were matched for comparability. As currently presented, baseline characteristics are 100-599 and 600-999, whereas outcomes are 20,100,200,400, and 1000.

Line 397: "In this current study, we found higher proportions of viral suppression in the

switch group when considering virologic failure thresholds other than <50 copies/mL, i.e. 20,

100, 200, and 400 copies/mL, but not for the current WHO recommended threshold of 1000

400 copies/mL."

- Is this simply because of time to suppression?

Line 446: "individual benefits need to be balanced against the public health implications.". 

- Individual benefits also need to be balanced against the individual harm of a higher frequency of side effects and worse adherence (which did not lead to observable worse outcomes in this trial but could lead to drug resistance later on)

*** Reviewer #2:

[see attachment] 

*** Reviewer #3: 

General comments

This is a highly relevant study which has been rigorously conducted in a setting representative of those were a majority of people living with HIV receive care. It addresses a common clinical problem in a randomized control trial. Although the occurrence of LLV during 1st line ART may decrease if dolutegravir replaces efavirenz as anchor drugs in sub-Saharan Africa, the findings of this study will still be of relevance, not least by highlighting and adding further evidence to the fact that attention must be paid to persons with detectable VL <1000 copies/ml. 

Major comments

Overall, I think the study is well performed and presented. I have no major comments on the paper. 

Minor comments 

1. In order to provide an estimate of the prevalence of LLV (according to the study definitions) in the uptake population, it would be interesting to present data from the total number of persons in the source cohort. In particular, it would be interesting to know how many had a VL in the range 100-999 copies/ml which was followed by VL<100 copies/ml (single measurement LLV). In the Hermans study from South Africa (ref 8), 23% had recorded VL in the range 51-999 copies/ml, but in most cases these were single measurement LLVs. The current study uses a stricter definition of LLV. Data on prevalence of repeat LLV in the study population could also affect the public health impact of regimen switch in LLV (as brought up in the discussion). Perhaps the first step is paying attention to VL in the LLV range, and not considering these as suppressed patients (as is currently the case) - and the proportion who have repeat LLV and need regimen switch is likely to be much lower. 

2. For the analysis, virological failure was defined as VL>50 copies/ml at 24 and 36 weeks. I recommend including a reference for this definition and/or a motivation of why this definition of virological failure was chosen. 

3. Methods, page 9, row 205: " … narrower pre-defined primary endpoint visit windows …" - I would appreciate an explanation of how this was done. 

4. Participants in the study were on long-term ART (median 5.9 years). A comment could be added in the discussion that the implications of the study may not applicable during the first 12 months after ART initiation. 

5. The levels of adherence are overall surprisingly low. This could reflect a more accurate recording of true adherence (often overrated), but may be commented in the discussion. 

6. Discussion, page 16-17: With regard to studies on regimen switch in patients with LLV, the authors could also mention a study by Boillat-Blanco et al (PMID: 24964403), with findings that support those of the current study. 

7. In the discussion, it would be interesting to mention different mechanisms of LLV, and how this could impact the effect of ART modification. For example, the risk of selection of drug resistance (and subsequent virological failure) is probably higher in persons with LLV due to ongoing viral replication during ART, and such individuals are likely to benefit from regimen switch. On the other hand, this may not be effective in cases where LLV is due to release of virions from latently infected cells (monoclonal LLVs). This is discussed in reference 23, showing that ART modification in LLV was effective in cases with demonstrable drug resistance and/or poor adherence, but not in cases in which these factors were not observed.

***

[LINK]

---

## [Editor Report · Decision Letter 2]

20 Jul 2020

Dear Dr. Labhardt,

Thank you very much for re-submitting your manuscript "Switch to second-line versus continued first-line antiretroviral therapy for patients with low-level HIV-1 viremia: an open-label randomized controlled trial in Lesotho" (PMEDICINE-D-20-02716R2) for consideration at PLOS Medicine.

I have discussed the paper with editorial colleagues and I am pleased to tell you that, provided the remaining editorial and production issues are dealt with, we expect to be able to accept the paper for publication in the journal.

[LINK]

Please let me know if you have any questions. Otherwise, we look forward to receiving the revised manuscript shortly. 

Sincerely,

Richard Turner, PhD

rturner@plos.org

Requests from Editors:

Please add a full competing interest declaration in the metadata, e.g., "All authors have declared that they have no competing interests.".

Please finalize the arrangements for data deposition. 

At line 43, please make that "... viral load (VL) measurements ..." or similar; and end the sentence on line 45 with "... second line treatment.".

We suggest adding a few words to the abstract to note that there were no admissions or deaths. Also, we suggest quoting the duration of ART.

At line 72, please make that "... in a higher proportion of participants with viral suppression". 

Please trim the author summary. For example, the first point can begin "WHO guidelines define ...". 

At line 101, please start the point with "We found that ..." or similar. 

Please remove trademarks from the text, e.g. at line 240. 

Please revisit the reference list, and add full access information where needed, e.g. to reference 35. 

***

[LINK]

---

## [Editor Report · Decision Letter 3]

11 Aug 2020

Dear Prof. Labhardt, 

On behalf of my colleagues and the academic editor, Dr. Nathan Ford, I am delighted to inform you that your manuscript entitled "Switch to second-line versus continued first-line antiretroviral therapy for patients with low-level HIV-1 viremia: an open-label randomized controlled trial in Lesotho" (PMEDICINE-D-20-02716R3) has been accepted for publication in PLOS Medicine. 

PRODUCTION PROCESS

PRESS

PROFILE INFORMATION

Thank you again for submitting the manuscript to PLOS Medicine. We look forward to publishing it. 

Best wishes, 

Richard Turner, PhD

Senior Editor 

PLOS Medicine

plosmedicine.org